# A Novel Low-Frequency Electromagnetic Active Inertial Sensor for Drug Detection

**DOI:** 10.3390/s24103059

**Published:** 2024-05-11

**Authors:** Erietta Vasilaki, Emmanouil Markoulakis, Diamanto Lazari, Antonia Psaroudaki, Ioannis Barbounakis, Emmanuel Antonidakis

**Affiliations:** 1Computer Technology, Informatics & Electronic Devices Lab, Department of Electronics Engineering, Hellenic Mediterranean University, Romanou 3, 73133 Chania, Greece; markoul@hmu.gr (E.M.); i.barbounakis@hmu.gr (I.B.); antonidakis@hmu.gr (E.A.); 2Department of Pharmacognosy-Pharmacology, School of Pharmacy, Faculty of Health Sciences, Aristotle University of Thessaloniki, University Campus, 54124 Thessaloniki, Greece; dlazari@pharm.auth.gr; 3Department of Nutrition and Dietetics Sciences, Hellenic Mediterranean University, Tripitos, 72300 Crete, Greece; psaroudaki@hmu.gr

**Keywords:** electromagnetic field, frequency, resonance, charging/excitation time, relaxation time, ULF, VLF, inertia sensor

## Abstract

The purpose of this paper is to demonstrate a new discovery regarding the interaction between materials and very low radio frequencies. Specifically, we observed a feedback response on an inertia active sensor when specific frequencies (around 2–4 kHz) are used to irradiate targeted pharmaceutical samples like aspirin or paracetamol drugs. The characteristics of this phenomenon, such as excitation and relaxation time, the relation between deceleration and a material’s quantity, and signal amplitude, are presented and analyzed. Although the underlying physics of this phenomenon is not yet known, we have shown that it has potential applications in remote identification of compounds, detection, and location sensing, as well as identifying substances that exist in plants without the need for any processing. This method is fast, accurate, low-cost, non-destructive, and non-invasive, making it a valuable area for further research that could yield spectacular results in the future.

## 1. Introduction

There are many existing methods that study the molecular structure, the components of a substance, or the existence of a substance in a material. These methods are called spectroscopic and are very useful both in chemistry and in other sciences, applied or not. Some of these methods, for example, are Nuclear Magnetic Resonance (NMR), IR Spectroscopy, Raman Spectroscopy, and Nuclear Quadrupole Resonance (NQR). The range of frequencies that are used varies from a few Hz to a few million Hz (GHz). The frequencies in use depend on the method we choose and on the level of study we want to deepen. For example, IR spectroscopy has a corresponding frequency range from 1.9 × 10^13^ to 1.2 × 10^14^ Hz and studies parts of molecules, such as functional groups, or the types of bonds that exist within a molecule. Additionally, each of these methods needs a different amount of time to give the appropriate results. It has been observed that the time a method needs to give a result depends on the kind of material we study, the field strength, and many other parameters.

In addition to the already existing methods, a new innovative method has been created in recent years and can detect a substance as a whole, as well as detect part of the molecule of this substance and find out which other substances include the same molecular part. This method uses frequencies between 1 Hz and 10 kHz, which belong to the band of ultra-low and very low frequencies, and although the natural phenomenon on which it is based is not known, it has been confirmed that it has accurate results.

Below is a brief description of the basic principles of each of the previous methods, emphasizing the time that is necessary for these methods to have an accurate result.

NMR uses an external magnetic field (B_0_) to stimulate the nuclei from their ground state to an excited state. When the field is applied, the spins of the atoms align in a parallel or antiparallel fashion. This creates a difference in their energy. The frequency that corresponds to this energy is called Larmor frequency. The larger the external field, the bigger the difference between the energy of the spins. The magnetic field that is used varies from 1 to 20 T [1,2], which corresponds to frequencies in the order of magnitude of MHz. An external rectangular radio frequency (RF) pulse is used as well. This pulse instantly reaches maximum. It stays at this level for a short duration (~10 μs) and then it instantly goes to zero. It is used to change the direction of the spins in order to be able to obtain information for the substance under study [3,4]. When the two fields are applied for a long time, the sample becomes saturated and a certain amount of time is needed to return to the initial state, the so-called relaxation time. There are two types of relaxation time. The first is T_1_, also known as the spin–lattice relaxation time. It is a measure of how quickly the net magnetization vector returns to its ground state in the direction of B_0_. The possible values of T_1_ range from a few tenths of a second to several seconds. The second relaxation time is spin–spin relaxation time, T_2_, and it is the time for the spins to lose coherence with one another. It can be equal to or less than T_1_. It varies from msec to sec [5,6]. With NMR, solid, liquid, aquatic (solutions), and gas [7,8] materials can be studied. In general, nuclei take longer to relax at higher fields, so more time is needed for the peaks to stand out from the noise that is created in each scan. An experiment with the NMR method is repeated multiple times so the signal-to-noise ratio is improved. This repetition of scans is predetermined by the user and could correspond to a huge number of scans; for example, 128 scans, 256 scans, 512 scans, or more, which equals a period of minutes, hours, or more.

Additionally, carbon atoms that are not creating direct bonds with hydrogen (protons) display much longer relaxation times than protonated carbons [9].

Samples like this have relaxation times that range between 5 and 30 min.

Another method that can identify and analyze the components of a substance is infrared (IR) spectroscopy. IR creates vibrations in molecules by using infrared radiation [10]. When a molecule absorbs infrared radiation, its bonds vibrate and create peaks in its spectrum. Each functional group or bond has peaks that correspond to different wavelengths. The frequencies that are used are in the order of Hz. IR is only used when a sample has a dipole moment; otherwise, it cannot be used [11,12,13]. IR is used to study and identify chemical substances or functional groups in solid, liquid, or gaseous forms. Powders can be studied as well. It cannot be used in aqueous solutions.

In addition, Raman spectroscopy is a non-invasive (non-destructive) technique that uses visible light [14]. The frequencies that are used are in the order of Hz. Lasers with monochromatic light are used in this method. When a high-intensity laser hits a molecule, the molecule scatters incident light. Most of this light is the same color (frequency) as the light of the laser. This is called the Rayleigh effect, and it does not provide any information about the molecule. A small amount of light, though, is scattered in different frequencies (colors), and with this scattering, information is received about chemical structure and molecular interactions. This is Raman scatter [15]. Raman spectroscopy is used when there is a change in polarizability, a change in the size, shape, or orientation of the electron cloud that surrounds the molecule. In the Raman spectrum, several peaks can be observed. Each peak takes place in a standard wavenumber, and it corresponds to a specific bond, such as C-C, C=C, N-O, C-H, etc. Raman spectroscopy is used to analyze solids; liquids; gels; gases; inorganic, organic, and biological materials; aqueous solutions; etc. [16,17].

Also, there is a method similar to NMR that is called Nuclear Quadrupole Resonance (NQR). The main difference between these two methods is that NQR does not need a strong external magnetic field. In general, the frequencies that take place in NQR are in the order of magnitude of kHz [18].

Nuclear Quadrupole Resonance (NQR) is another spectroscopic technique that is used to study the interactions between an electromagnetic field and the quadrupole moment of a nucleus. It is used in chemical elements, such as nitrogen, hydrogen, etc., which have non-zero magnetic quadrupole moments. It is applied in various fields, such as analyzing molecular structures and studying materials at the atomic level. The time that is needed to extract the results in NQR depends on various factors, such as the type of samples under study, the experimental parameters, etc. In general, simpler experiments need some hours to be completed, while more complex studies require longer acquisition times. It is common for more complex experiments to take several hours or even days in order to be completed. NQR uses radio frequencies between 10^4^ and 10^9^ Hz [19,20,21,22].

Lastly, another method used to detect materials or part of the components of some materials is a method that uses ultra-low (ULF) and very low frequencies (VLF). An inertial system moves around a fixed axis while being inclined. While the sensor rotates, its movement changes from accelerating to decelerating. When the sensor is activated, it emits electromagnetic radiation in the ULF and VLF frequency spectrum. The sensor can detect which materials are in the area around it when it emits the frequency of the specific materials. We place a material at a short distance in front of the detector and mark the position. The sensor generator is set to emit a certain frequency, which corresponds to the material we have placed in front of the detector. The device is allowed to move while emitting the frequency of the material. When the antenna of the sensor is aligned with the material, an external force is observed to act on the antenna, which slows its movement. This deceleration can be measured and indicates the existence of some material at the frequency we are emitting, as shown in Figure 1. In addition, it is observed that it takes a certain charging time until the sensor starts responding to the material and detecting it. After this initial charging time is achieved, a maximum inertia feedback response is observed in the motion of the sensor slowing it down [23,24,25]. It should be noted that it is a non-destructive and non-invasive method, which means that when we use a material, no changes happen to its composition, and it is not tampered with or altered by any means. Even though the physics behind this phenomenon is not known, it has been ascertained that the sensor works and has accurate results; therefore, this novel phenomenon is worthy of further research.

Next, we will describe the material and apparatus used in our experiment and also present and discuss the results of our research. More specifically, we will research the interaction between materials with ULF and VLF frequencies using an inertia active sensor.

## 2. Materials and Apparatus

To conduct the experiment, a novel sensor and medicinal substances that include common parts in their molecule were used.

These substances are paracetamol [26,27], mefenamic acid [28,29], acetylsalicylic acid [30,31,32], and ibuprofen [33]. Commercially available painkillers and anti-inflammatory drugs have been used to study the above substances.

In Table 1 below, we can see the chemical formulas of the painkillers that were used in the experiment.

Below is a brief description of the sensor that was used in conducting the experiment. The sensor is an inertia system that can move on a circular path. It consists of an electronic box, a telescopic antenna, a perpendicular axial fixed underneath the electronic box, and a base around which the sensor rotates. The electronic box contains a signal generator system that produces electromagnetic frequencies in the band of ULF and VLF (3 Hz–30 kHz). The antenna emits the produced signals. The system consists of two pieces, in general: the electronics box, which includes the signal generator system and has the antenna attached to it, and the perpendicular axial that is fixed underneath the box and allows the sensor to move in a circular movement. Under the axial, there is a base that supports the generator and the antenna. We put the inertia system at an initial position and allow it to move around its axis. The system starts its circular movement. In the first part of the movement, the sensor accelerates, while during the second part of its movement, it decelerates. The sensor is activated and starts emitting electromagnetic signals. The generator circuitry applies a signal to the antenna. Because different materials respond to different frequencies, we can apply different signals to the antenna depending on the type of material we wish to study. When the generator is switched off and no material is under study, the only force applied to the inertia system is the weight of the system. When the movement starts, friction between the electronic box and the perpendicular axial appears. An extra force, F, which is applied to the antenna of the inertia system and opposes the movement, appears when a material is located in a direction pointed to by the antenna. The electronics in the sensor box, as shown in Figure 2, are analyzed. The electronic circuitry consists of the signal generator circuitry, the force detection circuitry, and a wireless communication module. The signal generator circuitry has an LCD screen display where the user can see and select certain choices. It consists of a micro-computer unit (MCU) processor that receives commands from the user through a keyboard. Information about the produced signal for different kinds of materials can be stored in the MCU’s memory. The emitted frequencies are 1 Hz–30 kHz. The intensity of the signal amplitude can be adjusted up to 600 V RMS [34,35]. A 3.3 V lithium battery of 2700 mAh powers up the system. After the MCU receives data from the user, it gives a command to the Direct Digital Synthesis (DDS) Generator to produce the selected signal. The output of the DDS is amplified and sent to the antenna. The user can select the amplitude of the output signal through the screen display. The force detection circuitry consists of a vertical-axis accelerometer that is used as a vibration sensor. When an external force is applied to the antenna, it is observed that the inertia system vibrates mechanically with a specific frequency. The signal of the accelerometer is processed and then redirected into a second MCU. In this MCU, an FFT (Fast Fourier Transform) algorithm is applied to calculate the vibration frequencies. In the electronics box, there is a wireless communication module that transmits the angle that is detected to a computer where the data are collected. Finally, there is an angular positioning system through which the angles at which the antenna is oriented during its movement are precisely measured. The angles are calculated by using Gray code. Gray code, also known as reflected binary code, is an ordering of the binary numeral system where consecutive numbers differ in only one bit (binary digit). The Gray code disk is connected to a Raspberry Pi computer board, which sends the data to a computer. Through special software, we process these data and can present various diagrams.

## 3. Experimental Procedure

According to previous studies, it has been found that each substance corresponds to a specific frequency. This frequency detects part of the molecule of the substance; therefore, substances with a common part in their molecules can be detected at the same frequency.

For the substances that will be used for this specific experiment, the following centered corresponding frequencies [24] have been found, as shown in Table 2:2429 Hz, which is the common frequency for mefenamic acid and acetylsalicylic acid;2497 Hz, which is the common frequency of acetylsalicylic acid and ibuprofen;2390 Hz, which is the frequency of paracetamol.

### 3.1. Material Frequency Response

We place the sensor in a specific position where it can move around its axis in a circular movement. Radially around the sensor, at specific angles, we place the four substances to be studied. We activate the operation of the device and set it in motion. We then record the decelerations that appear on the antenna and send the data to the computer for processing. We make a diagram of deceleration in relation to the substances we use.

The purpose of this experiment is to show how materials are affected when they are irradiated with specific frequencies. More specifically, we prove that substances with common parts in their molecules are equally affected by the frequency at which they are tuned, while substances that do not have this frequency are affected less, or not at all, by this frequency.

The first frequency to be used is 2497 Hz, which is the corresponding frequency of ibuprofen and acetylsalicylic acid. In addition, a second diagram will be created for the frequency of 2429 Hz, which is common to mefenamic acid and acetylsalicylic acid.

### 3.2. Excitation Time vs. Type of Material

The sensor is placed in a prearranged position, around which it performs a free movement in which the acceleration changes constantly throughout the duration of the movement. In front of the sensor, at a certain angle, the material we want to study is placed. Then, we activate the sensor and program it to emit the frequency that corresponds to a certain material. We put the sensor in motion and record the data on the computer.

What interests us in this experiment is the charge–discharge time of the materials. More specifically, here we study the process of proving that there is a period of time that corresponds to the charging time, and at that time, the targeted material absorbs energy. An additional goal is to demonstrate that there is a maximum charging time—which is also called excitation time—in which the material resonates and confirm this through the experiment. Finally, it is important to see what happens after this phase when the material stops absorbing energy and, therefore, begins to discharge. This discharging occurs despite the continuous irradiation of the material by this active sensor.

This process will be repeated twice for the same quantities of two different substances. The first substance to be used is ibuprofen, which corresponds to the frequency of 2497 Hz, and the second substance to be used is paracetamol, which resonates to the frequency of 2390 Hz. The quantities of both substances are 40 g.

In the end, we present the inertia diagrams (deceleration versus time) measured as a function of time, a(t), for the above substances.

### 3.3. Charging Time vs. Material Quantity

The sensor is placed in a prearranged position. Two different types of substances and different amounts of each type are placed in front of the device in some predetermined position.

The device is activated and starts emitting a specific frequency. First, we use the specific quantities, mentioned below, of each substance and record the values of the deceleration versus time. Next, we increase the amount of the specific substance and record the new deceleration values. The same procedure is followed for a second substance.

This experiment aims to demonstrate whether the charging time depends on the quantity of the material we detect. In addition, we are interested in finding whether there is a dependence between the recorded deceleration and the amount of material detected.

To reach the necessary conclusions, we create deceleration vs. time diagrams per quantity of each of the two materials, paracetamol and ibuprofen.

The quantities of the substances that are going to be used are:40 g and 115 g of ibuprofen;40 g and 1060 g paracetamol.

### 3.4. Response to Sensor Signal Intensity

We place the sensor in a predetermined position again and, in front of it at a certain angle, we put the material to be studied. In this case, this substance is paracetamol, and the transmitted frequency corresponds to 2390 Hz.

What we are interested in in this particular part of the paper is to find out if there is a correlation between the charging time and the intensity of the signal amplitude of the frequency emitted by the sensor. To achieve this, we take multiple measurements for the values of the decelerations while changing the intensity each time.

Initially, 75% amplitude is used. Observed decelerations are recorded and a diagram of deceleration versus time is made. Then, the same procedure is repeated, progressively reducing the intensity each time, from 50% to 25% to 10%.

Our goal is to find out if the charging time is affected by the transmitted signal intensity. Additionally, we want to find out if charging times are inversely proportional to signal intensity coming from our active sensor.

Below, follow the results and an extended commutation on each phase of the previous experiments. The headline in each section remains the same as the headlines in the previous sections in order to be more obvious which paragraph of the previous chapter corresponds to the paragraphs of the following chapter.

### 3.5. Relaxation Time

It has been observed that the materials in this study absorb energy for a certain amount of time, the so-called excitation time. This amount of time depends on the signal amplitude. The lower the signal amplitude, the more time it takes for a substance to reach its excited level. However, when a substance reaches its excitation level, it cannot absorb any more energy. From this point and after, the substance does not absorb any more energy, although we continue to irradiate it with electromagnetic waves (E/M) of ULF and VLF frequencies. For the material to be able to absorb energy again, there must be a relaxation period where the material is not irradiated with E/M waves. This period of time is called relaxation time.

In this phase of the experiment, we record the decelerations of the sensor in three different moments in order to prove that when a material is saturated, then we cannot detect it anymore. Also, it is necessary to find out after how much time it will be possible to detect the material again. The first moment is right after the material is saturated. The second one is after 24 h have passed, and the third moment is after 48 h.

The material we used is paracetamol, while the signal amplitude is 100%.

## 4. Results and Discussion

### 4.1. Material Frequency Response

In the first phase of the experiment, we programmed the sensor to emit a frequency of 2497 Hz, which is the centered responding frequency of acetylsalicylic acid and ibuprofen. Since at this frequency, both the aforementioned materials are tuned, it is expected to observe the resonance phenomenon when the antenna aims at the two specific substances. This means that when the antenna aims at these materials, then a resonance feedback force, which is observed in all our experiments, is generated, and a maximum deceleration in the antenna motion is recorded. On the contrary, this is not the case when the antenna aims at the other two materials that are out of resonance.

The same principle applies to the second part of the experiment, where we use the substances mefenamic acid and acetylsalicylic acid. Both materials tune in to the frequency of 2429 Hz. So, we notice that in this case too, the substances that correspond to the frequency of 2429 Hz show the maximum deceleration, while the rest have a noticeably smaller one.

The above results can be shown in the following diagrams (Figure 3a,b). The first (a) corresponds to acetylsalicylic acid and ibuprofen for the frequency of 2497 Hz, while (b) corresponds to mefenamic acid and acetylsalicylic acid tuned to the frequency of 2429 Hz.

We notice that in both diagrams, the values of the decelerations of the substances that have a common frequency are exactly the same. For example, in the first diagram, acetylsalicylic acid and ibuprofen both exhibit a deceleration of 32 degrees/s^2^. Also, in the second diagram, both mefenamic acid and acetylsalicylic acid have the same deceleration value corresponding to 31 degrees/s^2^.

The maximum amplitude (100%) corresponds to 600 V. For this part of the experiment, we used a signal amplitude of 10% of the maximum amplitude, which is 60 V.

### 4.2. Excitation Time vs. Type of Material

It has been observed that as long as the transmitted frequency irradiates a material, this material charges until it reaches its maximum charge. This is called excitation time. As the charging of the material increases over time, the inertial feedback force observed on the antenna also increases, and the maximum value of deceleration is recorded. After the period of maximum charging ends, a period of time follows where the material cannot absorb any more energy, so it gradually discharges. This time corresponds to the relaxation time of the material. During this relaxation period (discharging) it is also evident that this deceleration effect diminishes.

These observations are proven based on the next diagram, Figure 4. In this diagram, we notice that for both ibuprofen and paracetamol, there is a charging period that corresponds to approximately 1440 s when the same amount of these two substances is used. At 1440 s, the resonance occurs, so we have the maximum deceleration for both substances, and then there is the discharge period where the deceleration decreases. Relaxation time is examined in the following section.

It is also important to note that the maximum charging time is independent of the substances used. Different substances of the same amount will obtain the maximum deceleration at the same period of time. Therefore, we can safely say that we cannot use this method to distinguish the substances solely by their recorded charging time. Although differences in the absolute amplitude values of deceleration between the two substances can be seen, they are not the primary focus and scope of this research. Our main interest lies in determining the time taken by the material sensors to achieve maximum deceleration (resonance) upon detection by the sensor.

It should be noted that we used 40 g of each substance, i.e., 40 g of paracetamol and 40 g of ibuprofen, and that the signal amplitude is the same as it was in the previous part of the experiment, that is, 10%.

### 4.3. Charging Time vs. Material Quantity

Observing the diagrams below, it is evident that the charging time is independent of the amount of material used. In the case of ibuprofen (Figure 5A), 40 g was used in the first measurement and 115 g in the second. In both cases, we notice that the deceleration achieves its maximum value at 1440 s. The same thing happens during the detection of paracetamol (Figure 5B), where 40 g and 1060 g of this substance are used. Both in the case of ibuprofen and in the case of paracetamol, regardless of quantity, the maximum charge, and therefore the maximum deceleration, is observed at 1440 s (24 min).

Nevertheless, based on the following diagrams, we notice that the maximum deceleration depends on the type of material. Different kinds of materials have different values of deceleration. Paracetamol has higher deceleration than ibuprofen (Figure 5A,B).

According to the two previous diagrams, it is obvious that there is no significant change in the time of charging relative to the quantity of the materials. All the materials gain their maximum charge at a fixed period of about 1440 s. Observing the two diagrams though, it is obvious that the maximum deceleration depends on the kind of material used. Paracetamol has a higher value of deceleration than ibuprofen. It has been observed in Figure 5 that the amount of material influences the deceleration to some extent. However, the correlation between the two magnitudes is currently not clear. Additional research is required to confirm this connection. Please note that the focus of this paper is solely on the effects of a material on charging time, and it does not explore its influence on deceleration.

The signal amplitude that was used was 10%.

### 4.4. Response to Sensor Signal Amplitude

In this phase of the experiment, we studied the relation of the materials’ charging time with the amplitude of the emitted signal from the active sensor. In contrast to the previous experiments mentioned, this time we observed a direct dependence of these two quantities, signal strength versus charging time.

More specifically, it was observed that as the intensity of the emitted radiation increases, it causes the charging time to decrease. From repeated measurements, however, we observed that the two characteristics we measure, signal strength versus charging time, are inversely proportional (not linear), as shown in Figure 6. For this specific part of the experiment, 1060 g of paracetamol was used.

So, we observe based on Figure 6 the direct dependence of the charging time on the intensity of the radiation emitted by the sensor. More specifically, as the intensity—i.e., the amplitude—emitted by the sensor increases, the time needed to reach the maximum charge decreases.

We notice that the maximum charge is reached at 1500 s (25 min) when the amplitude is 10% (Figure 6a). In the case that the sensor emits radiation of 25% intensity, then the maximum charging time corresponds to 1000 s (17 min) (Figure 6b). In Figure 6c, we see that the maximum charging time occurs at 700 s (12 min) when the intensity of the transmitted frequency is 50%. Finally, we notice based on Figure 6d that when the amplitude is 75%, the maximum charging time corresponds to 600 s (10 min).

It should be noted that in the diagrams, we indicate the absolute values of measured decelerations.

### 4.5. Relaxation Time

After multiple experiments, it is shown that the relaxation time is about 48 h. If we try to repeat the same experiment in less than 48 h, we observe that the sensor cannot detect the materials under study. It seems that the materials are still “saturated”/charged, so they cannot absorb more energy, and for that reason, there cannot be seen an observable deceleration. When the materials are saturated, we cannot observe the gradually increasing deceleration. We notice that in the angle where the material is placed, the decelerations have low values that remain almost the same and do not increase; therefore, there is not a distinctive peak. This means that the sensor does not detect the materials under study due to the over-charged state of the material. As can be seen in Figure 7a, when we try to detect the material instantly after it becomes saturated, the deceleration has values between 10 degrees/s^2^ and 12 degrees/s^2^, and it remains at the same values throughout the whole duration of the experiment without any peak standing out. When we repeat the same procedure after 24 h of the saturation of the material, it can be seen that there is still not a distinctive peak; however, the values of the deceleration range between 6 degrees/s^2^ and 9 degrees/s^2^, and therefore, there is a bigger fluctuation between the values of the deceleration (Figure 7b). This means that the material is not at its full charge anymore and can absorb a small amount of the energy we irradiate. Finally, we repeat the same process after 48 h, which is an adequate period of time for the material to absorb the energy with which we irradiate it. As we can observe in Figure 7c, the deceleration increases progressively and reaches its maximum value at 400 s. We can safely assume then that 400 s is the excitation time and at that moment the sensor detects the material under study. In conclusion, it can be said that the relaxation time for this method corresponds to 48 h. To reach the saturation point, the sensor and the material system require irradiation for an extended period of over an hour. However, this is an extreme scenario as the sensor typically does not need that long to detect the material. The reason for the long irradiation period was to observe the sensor’s behavior when emitting for an extended period.

It should be noted that the signal amplitude that was used in this section was 100% [35].

## 5. Summary

It has been observed that when a material is exposed to very low and ultra-low electromagnetic radiation, there is an interaction between the material and the frequencies used. This interaction results in the material becoming charged and resonating with the sensor that emits the specific frequency. During this resonance, a force is exerted on the antenna of the sensor, and as a result, we can detect the existence of the material in the direction of the sensor. This is a novel spectroscopic method that can be used for identifying and detecting substances. It can detect both small and large amounts of substances, making it incredibly useful when substances are present in small quantities. Additionally, this method can identify a substance without processing, which means that samples are not destroyed or altered. This is particularly useful when the materials are rare or expensive. Moreover, since this method can identify a substance without processing, it could be used to study unknown samples for the existence of certain substances. In the future, this method could be used in the field of pharmacology to identify if a drug contains a certain substance. It could also be used to detect and identify substances that contribute significantly to our health, such as secondary metabolites found in herbs and edible plants. Furthermore, this method could be used for detecting explosives and illegal and dangerous materials, as well as the identification and detection of illegal drugs. In conclusion, this method would be useful in many areas for multiple purposes, and it is worthy of further research.

## Figures and Tables

**Figure 1 sensors-24-03059-f001:**
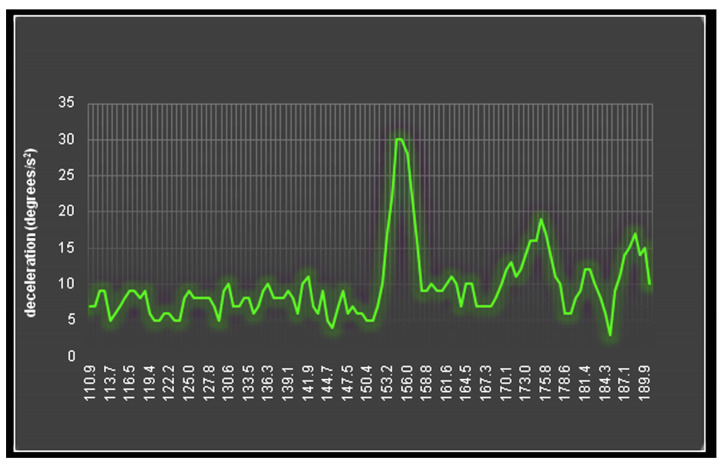
Deceleration in each angle for the innovative method. When the antenna aims at the material in study, a maximum force is applied to the antenna. This force corresponds to the maximum deceleration.

**Figure 2 sensors-24-03059-f002:**
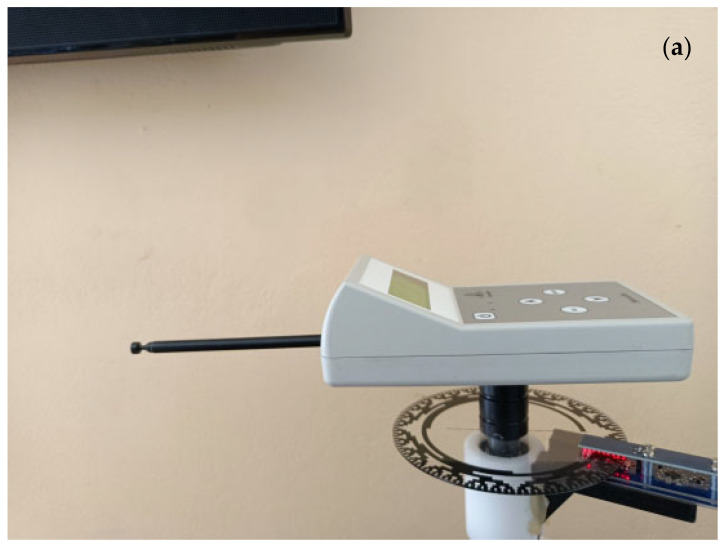
(**a**) Inertia system sensor, with Gray code disk and data transmission to a computer. It rotates in a circular movement around a vertical shaft. The sensor emits ultra-low (ULF) and very low frequencies (VLF) (3 Hz–30 kHz). (**b**) Block diagram of the main structure of the experimental layout.

**Figure 3 sensors-24-03059-f003:**
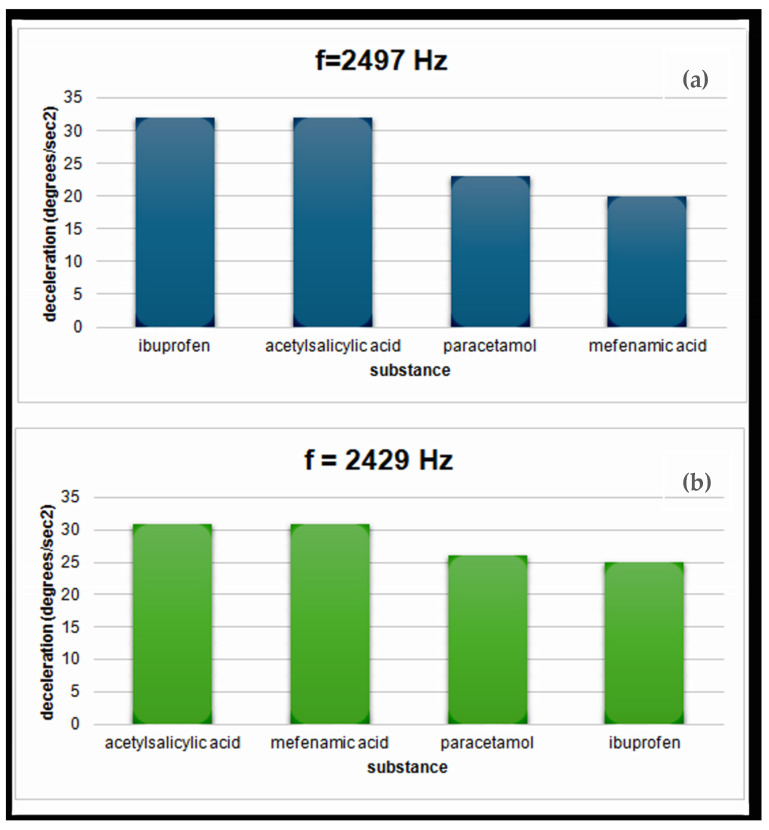
Deceleration graph as a function of material type. The diagram shows how a material is affected by a transmitted frequency. Materials with common parts are maximally affected. (**a**) Response frequency for ibuprofen and acetylsalicylic acid at 2497 Hz; (**b**) response frequency for acetylsalicylic acid and mefenamic acid at 2429 Hz.

**Figure 4 sensors-24-03059-f004:**
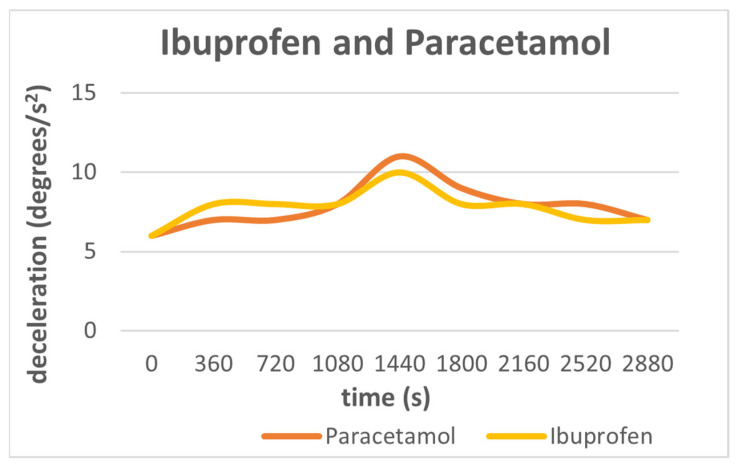
Deceleration versus time for ibuprofen (yellow) for the frequency of 2497 Hz and paracetamol (orange) for the frequency of 2390 Hz for 40 g of each substance. It can be observed that charging time is independent of the material we study. For both substances, the excitation time is at 1440 s.

**Figure 5 sensors-24-03059-f005:**
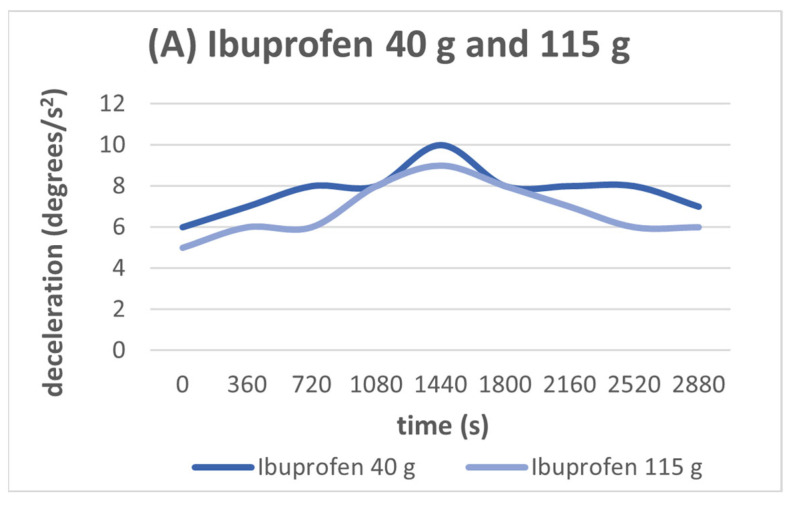
(**A**) Deceleration versus time for 40 g of ibuprofen and 115 g of ibuprofen for the frequency of 2497 Hz. (**B**) Deceleration versus time for 40 g of paracetamol and 1060 g paracetamol for the frequency of 2390 Hz. We observe that charging time does not depend on the quantity of the material in use.

**Figure 6 sensors-24-03059-f006:**
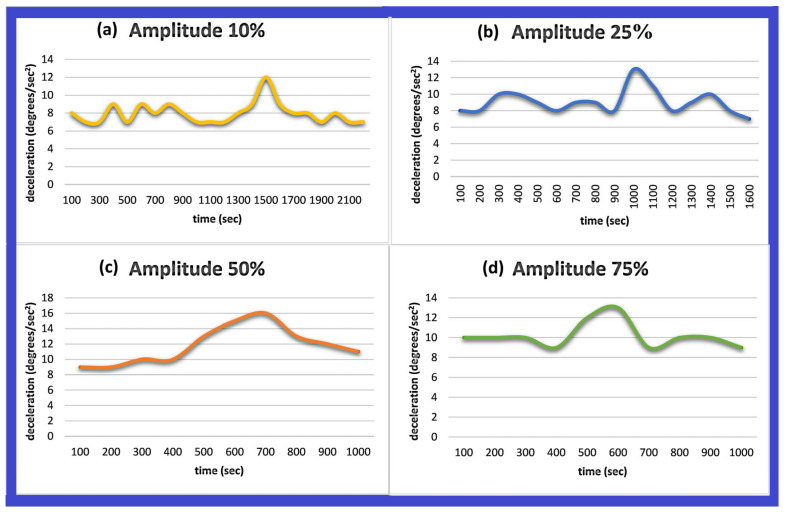
Different cases of deceleration. Diagrams of time vs. amplitude, while the quantity of the material remains the same. It can be seen that when the signal amplitude increases, then the time for paracetamol to obtain the maximum charge decreases. The amplitudes that were used for this experiment were 10%, 25%, 50%, and 75%.

**Figure 7 sensors-24-03059-f007:**
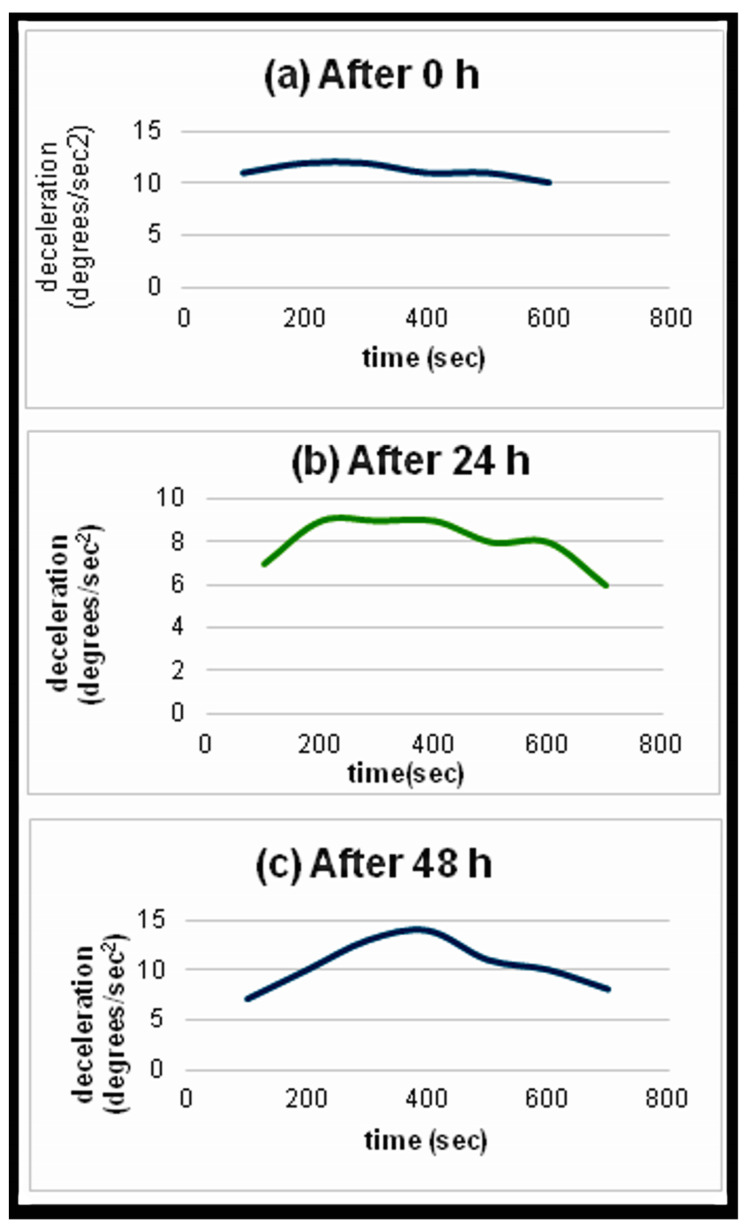
(**a**) Irradiating the charged material 0 h after its saturation. The material cannot absorb any amount of energy; therefore, the decelerations do not change significantly. (**b**) Trying to detect the material 24 h after its saturation. There can be seen a small increase in the inclination of the chart, which means that the material was able to absorb a small amount of energy. (**c**) Waiting 48 h before we irradiate the material. The material is fully discharged; therefore, it can absorb the maximum amount of energy in order for the novel phenomenon to take place. The deceleration has a distinctive peak at 400 s, which is the excitation time of the material when the signal amplitude is 100%.

**Table 1 sensors-24-03059-t001:** Chemical substances used in the experiments.

Name	Chemical Formula	3D Simulation
acetylsalicylic acid	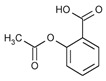	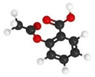
mefenamic acid	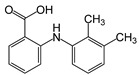	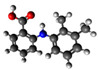
paracetamol	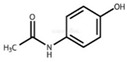	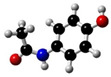
ibuprofen	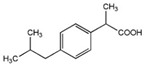	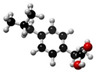

**Table 2 sensors-24-03059-t002:** Centered detection frequency found by the experiment for each test sample.

Frequency (Hz)	Acetylsalicylic Acid	Paracetamol	Ibuprofen	Mefenamic Acid
2390		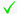		
2429	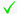			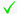
2497	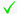		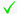	

## Data Availability

Data are contained within the article.

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
