# Peer review of "A Novel Low-Frequency Electromagnetic Active Inertial Sensor for Drug Detection"

_sensors, 2024, doi:10.3390/s24103059_

Round 1
Reviewer 1 Report
Comments and Suggestions for Authors
In this study, feedback responses on an inertial active sensor are reported when a remotely targeted drug sample, such as an aspirin or paracetamol drug, is irradiated at a non-ionizing very low radio frequency (VLF) of around 2-4 kHz. Each compound resonates to a specific resonant fundamental frequency within the above range. This newly observed phenomenon may have applications in the fields of remote identification, detection, and location sensing of compounds. I think it can be published on Sensors after revisions:
(1) Please annotate the information corresponding to Figure 4 in the article. The author should accurately mark Figure a and b in Figure 7.
(2) In the material frequency response experiment section, the author explained that a signal amplitude of 10 % of the maximum amplitude (600V) was used, but the author explained below that 600V was used. Please the author explain.
(3) Please explain in detail the changes in deceleration of the two substances and explain their differences according to the information in Figure 8. The color of the curve in the figure note in Figure 8 is incorrectly interpreted, and the authors are requested to correct it. add some references such as “Carbon, 2024, 221: 118925., Small, 2024, 20: 2305849., J Mater Sci Technol, 2024, 189: 155-165., Nano-Micro Letters, 2024, 16(1): 6., doi: 10.1007/s12613-024-2875-y.”
(4) It can be seen in Figure 9 that the amount of material has a certain influence on the deceleration, please explain in detail. In section 4.4, the author mentions the relationship between the charging time and the intensity of the sensor emission observed in Figure 8, which is inconsistent with the information in the figure. Please explain.
(5) According to the information in Figure 10, please further explain the relationship between emitted radiation intensity and charging time. In Figure 11, it can be seen that the material exhibits fluctuations during testing after saturation, and the fluctuations detected at 0 h, 24 h, and 48 h are different. Please explain. Please pay attention to the sentence grammar and picture quality in the article.
Comments on the Quality of English LanguageMinor editing of English language required
Author Response
Please see the attachment: Response letter to reviewer 1.

Reviewer 2 Report
Comments and Suggestions for Authors
The authors have provided a possible interesting paper. Nevertheless, I think that they must explain the practical advantages of their findings in daily practice, also in clinical pharmacology setting.
Figure's size (especially figure 1 and 2) must be increased since they are difficult to read.
Furthermore, drugs name must not be written using capital letters.
Author Response
Please see the attachment: Response letter to Reviewer 2.

Reviewer 3 Report
Comments and Suggestions for Authors
The authors studied a novel electromagnetic VLF active inertial sensor for drug detection. This experiment aims to show how materials are affected when irradiating with specific frequencies.
Please increase or improve the size and resolution of Figures 1, 2, 3, and 4.
Correction in Figure 8: The colors are used because they differ from the figure caption's identification.
Author Response
Please see the attachment: Response letter to Reviewer 3.

Reviewer 4 Report
Comments and Suggestions for Authors
The authors propose a method for drug detection using inertial sensors. This is a useful endeavor that could have potential applications in the field of remote identification, detection and location sensing of compounds. However, this manuscript still has some shortcomings, as follows:
1. The abbreviation VLF is given when the text first appears, so it is not appropriate for this abbreviation to appear in the title.
2. Abstract is not concise enough, it does not grasp the core and outstanding highlights. In fact, this manuscript uses inertial active sensors to study the interaction between materials and ultra-low frequency and ultra-low frequency, and has achieved the purpose of identification and drug detection.
3. In the introduction of academic papers, usually do not appear a lot of figures. There are 5 figures in this introduction, which is too many even for review papers or popular science articles. In fact, a figure is not necessary to introduce each method.
4. Fig.6 shows the inertial sensor and its practical application, which cannot reflect the test principle. I suggest adding a diagram of the testing principles in Fig.6.
5. Please explain the interaction mechanism between sensors and physical objects. It has been shown that the forces between materials are closely related to their space charge and equivalent dipole moment [Frontiers of Physics 18(2023)43302].
6. In the specific test, the material to be tested needs to be placed in a specific location. For unknown materials, how to choose this location? Please explain further.
7. Conclusion is not a summary of each part, it should be a highly refined research results. This conclusion is rather complicated, and it is suggested that the authors should simplify it.
Author Response
Please see the attachment: Response letter to reviewer 4.

Round 2
Reviewer 4 Report
Comments and Suggestions for Authors
I have no further comment.